# The Beak of Eukaryotic Ribosomes: Life, Work and Miracles

**DOI:** 10.3390/biom14070882

**Published:** 2024-07-22

**Authors:** Sara Martín-Villanueva, Carla V. Galmozzi, Carmen Ruger-Herreros, Dieter Kressler, Jesús de la Cruz

**Affiliations:** 1Instituto de Biomedicina de Sevilla, Hospital Universitario Virgen del Rocío/CSIC/Universidad de Sevilla, E-41013 Seville, Spain; smartin9@us.es (S.M.-V.); cgalmozzi@us.es (C.V.G.); carmenruger@us.es (C.R.-H.); 2Departamento de Genética, Facultad de Biología, Universidad de Sevilla, E-41012 Seville, Spain; 3Department of Biology, University of Fribourg, CH-1700 Fribourg, Switzerland; dieter.kressler@unifr.ch

**Keywords:** ribosomal protuberances, eS12, eS31, Ubi3, eS10, translation accuracy, ribosome biogenesis

## Abstract

Ribosomes are not totally globular machines. Instead, they comprise prominent structural protrusions and a myriad of tentacle-like projections, which are frequently made up of ribosomal RNA expansion segments and N- or C-terminal extensions of ribosomal proteins. This is more evident in higher eukaryotic ribosomes. One of the most characteristic protrusions, present in small ribosomal subunits in all three domains of life, is the so-called beak, which is relevant for the function and regulation of the ribosome’s activities. During evolution, the beak has transitioned from an all ribosomal RNA structure (helix h33 in 16S rRNA) in bacteria, to an arrangement formed by three ribosomal proteins, eS10, eS12 and eS31, and a smaller h33 ribosomal RNA in eukaryotes. In this review, we describe the different structural and functional properties of the eukaryotic beak. We discuss the state-of-the-art concerning its composition and functional significance, including other processes apparently not related to translation, and the dynamics of its assembly in yeast and human cells. Moreover, we outline the current view about the relevance of the beak’s components in human diseases, especially in ribosomopathies and cancer.

## 1. Introduction: Protuberances of the Ribosomal Subunits

Ribosomes are intricate molecular machinery found in the cytoplasm of all organisms. They play crucial roles during the translation of cellular mRNAs by efficiently and accurately decoding their genetic information. In addition, ribosomes can be found inside two types of organelles, chloroplasts and mitochondria [1,2,3,4].

All ribosomes are universally composed of two subunits, the small and large ribosomal subunits (r-subunits), which comprise ribosomal RNAs (rRNAs) and ribosomal proteins (r-proteins). The large r-subunit is about twice the size of the small r-subunit [2,5]. In a model eukaryote, such as the yeast *Saccharomyces cerevisiae*, the small r-subunit or 40S is composed of a single 18S rRNA and 33 different r-proteins; in turn, the large r-subunit or 60S contains three rRNAs (5S, 5.8S and 25S rRNAs) and 46 (47 in humans) r-proteins [6,7]. The general shape of the ribosomes has been known since the early 1970s; however, structures at high resolution have only been obtained since the beginning of the twenty-first century, following significant advances in structural technologies, such as X-ray crystallography and cryo-electron microscopy (cryo-EM) (e.g., [7,8,9,10]). In agreement with the functional conservation of ribosomes in all organisms [2,5,11,12], ribosomes display considerable structural similarity in prokaryotes, organelles and eukaryotes, although organellar ribosomes have substantially diverged from bacterial ones, with which they share a common ancestor (e.g., [3,13]). Moreover, structural studies have clearly shown that eukaryotic ribosomes (25–30 nm in diameter) are larger and more complex than their prokaryotic (20 nm in diameter) and organellar counterparts [2,3,5,6,7]. When observed at low resolution, ribosomes appear flattened and spherical in shape; however, this observation is far from being correct. In fact, both r-subunits are extremely irregular complexes containing different domains and exhibiting different discernible protuberances. The small r-subunit is formed by four different structural domains, the head, the platform, the body and the beak; the beak domain is the main protrusion found in small r-subunits (Figure 1A). The large r-subunit, which is a more compact unit, displays three characteristic protuberances, the L1- and the P-stalk and the central protuberance (Figure 1B). Importantly, the structure of several r-subunit protuberances and protrusions has only been determined at high resolution in a few organisms, a fact that is most likely due to the high mobility of these regions, which is clearly connected to their functions (e.g., [6]). Thus, the movement of the beak as part of the head of 40S r-subunits is important to allow the loading of mRNA on the ribosome and the interaction with translation factors, as we will discuss in detail later on. In turn, the primary role of the P-stalk is to provide a flexible hook outside of the core of the ribosome to recruit and then stimulate the activity of different translational GTPases during various stages of translation (e.g., [14,15,16]). The P-stalk also enables the activation of the Gcn2 kinase and mediates the interaction with distinct ribosome-inactivating proteins (e.g., [17,18,19,20]). The L1-stalk acts as a flexible protuberance at the antipodes of the P-stalk, facilitating the binding, movement and release of the deacylated tRNAs [21,22].

In addition to being part of the prominent, above-described protuberances, many r-proteins have long non-globular extensions, typically serpentine ones that become deeply embedded inside the rRNA core of the r-subunits and stabilize their overall structure. There are also r-protein extensions (see Figure 1B for examples) that project out of the ribosome and may serve as additional regions for interactions of the ribosome with factors or cellular structures [23,24,25]. Frequently, these extensions, which are located at the N- and/or C-terminal ends of r-proteins, are not well ordered and are highly mobile due to the lack of stable interactions with the core of the ribosome; consequently, it has not been possible to model all of them from the current cryo-EM density maps.

Protuberances are not only flexible components of ribosomes, but also dynamic structures. The eukaryotic P-stalk is a pentameric complex, composed of the essential r-protein uL10 (previously named P0) and two non-essential heterodimers made up of another two r-proteins (named P1 and P2), that is connected to the body of the 60S r-subunit by the uL11 r-protein [15,26]. P1 and P2, when phosphorylated, can exchange during translation with their equivalent cytoplasmic pool of non-phosphorylated P1 and P2 variants [27,28,29]. Interestingly, in yeast, P1 and P2 r-proteins are not essential for cell viability, but their absence affects differently the translation of specific mRNAs [15,30]. Furthermore, it has been described that the amount of P1/P2 r-proteins bound to the ribosome changes under different physiological conditions. For instance, when yeast cells enter the stationary phase, the P1/P2 r-proteins are practically absent from ribosomes [31]. This and other observations made in different eukaryotes (e.g., [32,33]) have allowed different authors to propose that the eukaryotic P-stalk acts as a regulator of translation and that changes in its composition could selectively control the translation of specific groups of mRNAs [34]. In agreement with a regulatory role of the P-stalk during translation, mutation of the phosphorylation sites of the yeast acidic P1/P2 r-proteins does not alter their interaction with the ribosome but influences the translation of specific mRNAs, among them, those related to osmotic stress [35]. Further experiments, using high-resolution techniques (e.g., ribosome profiling), are required to provide details on such a regulatory process and on the specific proteome translated upon changes in P-stalk abundance and phosphorylation status.

To the best of our knowledge, aside from the acidic P1/P2 r-proteins, there is no evidence of exchangeability of r-proteins from the other ribosomal protuberances, including the beak; however, there are interesting reports about several other r-proteins whose nascent forms can replace an older copy of themselves on a mature ribosome. (i) Thus, uL16 has been described as an r-protein potentially able to cycle on and off large r-subunits [36,37,38]. uL16 is strategically positioned on the surface of the evolutionarily conserved core of the 60S r-subunit, near the corridor through which aminoacyl-tRNAs move during accommodation and also near other functional centers, such as the GTPase-associated center (GAC) and the peptidyl transferase center (PTC). uL16 is required for the joining of r-subunits during translation initiation and the rotation status of the ribosome [39,40,41]; importantly, the availability to assemble uL16 onto large r-subunits could also be used as a translational regulatory mechanism to limit global translation under unfavorable circumstances [37,41]. (ii) Another interesting r-protein is RACK1 (Asc1 in yeast). RACK1 is a WD40-domain protein located at the head region of the 40S r-subunit near the mRNA exit site, where it interacts with other r-proteins, among them uS3, several kinases, and translation initiation factors [42,43]. RACK1 has an important role in different aspects of the translation process: it is required for efficient translation of mRNAs with short open reading frames (e.g., those of r-protein genes), it is critical for translation during heat stress, and it facilitates ribosome-associated quality control (RQC) mechanisms such as those that are involved in the rescue of stalled collided ribosomes at consecutive rare codons, such as CGA (Arg) in yeast [44,45,46,47,48,49]. Although there is no experimental demonstration for RACK1 to cycle on and off ribosomes [50], it is clear that RACK1 is a non-essential r-protein whose loss does not disrupt ribosome integrity and translation [46]. Moreover, as RACK1 protein levels can be modulated by a variety of environmental insults, such as hypoxic stress, glucose deprivation or amino acid starvation or by the physiological cellular status (exponential versus stationary growth phase) (e.g., [51]), it is possible that the RACK1’s ribosome association could be regulated, thereby, promoting differential translation. (iii) Another important inductor of exchangeability is chemical damage. Originally reported in prokaryotes [52], yeast ribosomes containing chemically damaged r-proteins can also be repaired by exchanging these with undamaged r-proteins, as convincingly demonstrated by the Karbstein laboratory [38,53,54]. The repair of damaged ribosomes might represent an important mechanism for maintaining the translational activity of cells following different insults as, for example, those produced by oxidative stress [53,55]. An analogous repair process via protein replacement has also been suggested to occur in neurons; thus, implying that this mechanism has been evolutionarily conserved (e.g., [56]). In yeast, the molecular details for a ribosome repair mechanism have been provided upon oxidation of eS26 and uL16, which are released from damaged ribosomes by their respective dedicated chaperones, Tsr2 and Sqt1, generating transiently eS26- and uL16-deficient ribosomes that are subsequently repaired with newly made r-proteins [38]. Ribosomes lacking eS26 can also be generated in a Tsr2-dependent manner upon the exposure of yeast cells to high salt or high pH conditions. Ribosome repair is extremely relevant from the physiological point of view as eS26-lacking ribosomes preferentially translate specific transcripts bearing Kozak sequence variations, including mRNAs enabling the biological response to high salt and high pH insults [55]. The recovery from stress is concomitant to the reincorporation of eS26 into ribosomes, again in a Tsr2-dependent manner [38,53]. This sophisticated system of autoregulation resembles that previously reported for the translation circuit of leaderless mRNAs (lmRNAs) in bacteria. These lmRNAs can be generated in response to adverse environmental conditions, some of them (e.g., the presence of antibiotics such as kasugamycin in the culture media) also being able to reprogram ribosomes to translate preferentially lmRNAs. Interestingly, this reprogramming involves the formation of stable r-particles (referred to as 61S particles) deficient in almost a dozen r-proteins from the small r-subunit, among them bS1 and other r-proteins associated along the path of the mRNA through this r-subunit [57,58].

Of special attention is the central protuberance (CP), where the 5S ribonucleoprotein particle (RNP), which is composed of 5S rRNA and r-proteins uL5 and uL18, plays an important regulatory role (Figure 1B). The whole 5S RNP, rather than its individual components, is incorporated as a prefabricated complex into early pre-60S r-particles during the nucleolar ribosome biogenesis phase, and it temporally adopts a conformation that is different from the one in the mature 60S r-subunit (e.g., [59,60,61]). From yeast to humans, ribosome biogenesis is tightly coupled to cell growth and proliferation, with the assembly of the 5S RNP playing a central regulatory role. Thus, in yeast, an imbalanced production of rRNAs and r-proteins generates defects in ribosome biogenesis leading to the accumulation of ribosome-unbound uL18, likely as part of the 5S RNP, which induces a delay in the G1/S phase of the cell cycle [62]. This behavior has been interpreted as part of a protective mechanism that prevents cell cycle progression when ribosome biogenesis is impaired, i.e., when not all necessary components are sufficiently available to ensure a complete and satisfactory assembly of ribosomes. In metazoans, the 5S RNP clearly accumulates when ribosome biogenesis is impaired [63,64,65,66]. The free 5S RNP binds to MDM2 (HDM2 in humans), which is an E3 ubiquitin ligase that ubiquitinates p53 and thereby channels it to degradation via the proteasome. Upon binding of the 5S RNP to MDM2, which occurs mutually exclusively to its binding to pre-60S r-particles, p53 escapes from MDM2-mediated degradation and accumulates [67,68]. Concomitantly, p53 is activated and, thus, exerts its different anti-proliferative functions, ranging from temporary cell cycle arrest to apoptosis [69]. The implications of the involvement of ribosome biogenesis in the regulation of p53 in human health and disease have been extensively discussed in other reviews (e.g., [68,70,71,72]) and will also be examined later in this work.

This review is aimed at giving insights into the composition, structure, role, biogenesis and dysfunction of the components of the beak, which is the most prominent protuberance found in all small r-subunits and so-called because of its resemblance to a bird’s beak. We discuss all these features of the eukaryotic beak by specially focusing on the current knowledge about the beak of 40S r-subunits from the yeast *S. cerevisiae* and by highlighting similarities and differences compared to the beak of human ribosomes.

## 2. Composition of the Beak of the Eukaryotic Ribosome

Despite the fact that the beak is overall an easily recognizable structure in the small r-subunits of all ribosomes, the composition of the beak is quite different within the three domains of life. In bacteria, the beak is composed exclusively of rRNA, specifically, the helix h33 of 16S rRNA [9,73,74]. In contrast, the beak of eukaryotic ribosomes has been transformed into a mixture of rRNA and specific r-proteins not found in bacteria, with the biological reasons for this transformation remaining unsolved. The r-proteins bound to helix h33 of eukaryotic 18S rRNA, which itself is shorter than the bacterial h33, are eS10, eS12 and eS31 [1,2,6]. It is accepted that the structural core components of the archaeal ribosomes are of prokaryotic origin, to which specific elements, some shared with eukaryotes, have been added; therefore, archaeal ribosomes represent intermediate steps towards the evolution of eukaryotic ribosomes [75]. In this sense, it is interesting to mention that the beak of archaeal small r-subunits has a transitional complexity from an all-rRNA to an rRNA/r-protein protrusion; accordingly, many archaeal genomes encode clear homologs of the eukaryotic eS31 r-protein, which in all cases, however, lack the eukaryote-specific N-terminal extension [76,77]. In addition, cryo-EM has shown that ribosomes of distinct archaea contain at least two copies of eL8, one at the canonical location on the large r-subunit and another one bound at a position on h33 that is equivalent to the one occupied by eS12 on the eukaryotic beak [78]; further predictions suggest that this feature, i.e., the presence of eL8 in the beak, occurs in all archaeal ribosomes [78]. Moreover, this observation suggests that eS12 evolved from eL8, as both proteins share conserved regions and belong to the same family (InterPro entry IPR004038). Finally, it is clear that eukaryotic eS10 has no counterpart in archaeal ribosomes, as evidenced by different database searches, including BLAST [77,79]. A comparison of the beak composition and structure of ribosomes from prototypical bacteria, archaea and eukaryotes is shown in Figure 2.

Ribosomes present in organelles also contain all the structural landmarks that are characteristic of cytoplasmic ribosomes of prokaryotes and eukaryotes, including the beak. However, mito- and chlororibosomes have been found to be extremely diverse in terms of their composition, including the acquisition of organelle-specific r-proteins that has an impact on their overall structures. In general, chlororibosomes resemble bacterial ribosomes [80,81,82]; hence, the beak of these ribosomes is formed by the h33 rRNA protrusion and is devoid of r-proteins (Figure 3). In marked contrast to chlororibosomes, the characterization of mitoribosomes from diverse species representing the different major groups of eukaryotes has revealed that these have diverged considerably from each other and from prokaryotic and eukaryotic ribosomes [3,83,84]. In mitoribosomes, the beak can vary from the all-RNA prototype found in yeast to the massive protein-based beak found in the kinetoplastid *Trypanosoma brucei* (Figure 3).

Cytoplasmic ribosomes of the yeast *S. cerevisiae* contain a standard beak, composed of a helix h33 of 52 nucleotides and a single copy of three r-proteins, eS10, eS12 and eS31 (Figure 4) [6]. Yeast eS10 is encoded by two paralogous genes, *RPS10A* (YOR293W) and *RPS10B* (YMR230W). The two genes code for the virtually identical eS10A and eS10B r-proteins of 105 amino acids and ca. 12.7 kDa that only differ in three solvent-exposed amino acids (E6, D7 and T98 in eS10A versus Q6, E7 and S98 in eS10B). Mutants harboring individual deletions of the *RPS10A* and *RPS10B* genes are viable in different yeast backgrounds; moreover, while the *rps10B∆* null mutant grows practically identical to the wild-type strain, the *rps10A∆* null mutant exhibits only a mild increase in the doubling time [85]. In all genetic backgrounds, eS10 is an essential protein as the *rps10A∆ rps10B∆* double mutant is inviable [85,86]. Yeast eS10 is a mostly globular protein; however, it contains an unstructured C-terminal extension of about 20 amino acids, which interacts with uS3 at the base of the beak in the mRNA entry channel [87]. In contrast to eS10 and most yeast r-proteins, eS12 and eS31 are non-essential r-proteins that are encoded by single-copy genes (*RPS12* or YOR369C, and *RPS31* or *UBI3* or YLR167W, respectively). However, in most genetic backgrounds, both the *rps12∆* and the *ubi3∆* mutant display a severe growth impairment [85,88,89]. Yeast eS12 is a small globular protein of 143 amino acids and ca. 15.5 kDa, containing an unstructured N-terminal extension of around 25 amino acids whose deletion causes a slow growth phenotype of still uncertain significance (our unpublished results). On the other hand, yeast eS31 is a small r-protein of 76 amino acids consisting of a globular domain, which is well conserved from archaea to eukaryotes, and a eukaryote-specific N-terminal extension of about 25 amino acids, which extends toward the ribosomal A-site and has relevant functions in translation and small r-subunit assembly ([1,76,90]; see later). More interestingly, it has been well reported that in most eukaryotes eS31 as well as eL40 are produced as C-terminal parts of ubiquitin-fused precursor proteins, which are rapidly processed to individual ubiquitin and r-protein moieties before assembly of the corresponding r-protein into the small and large r-subunit, respectively [91]. The biological relevance of maintaining these fusions during evolution for the correct production and assembly of these r-proteins and for the possible co-regulation of two related cellular functions, protein synthesis and protein degradation, has been previously covered and will not be further discussed in this review [91,92].

In consonance with the critical importance of yeast eS10, eS12 and eS31 r-proteins for cell growth, it has also been reported that loss-of-function mutations in the genes coding for these proteins in other model eukaryotes (*Caenorhabditis elegans*, *Drosophila melanogaster*, *Danio rerio*, *Mus musculus*, *Homo sapiens*) lead to a myriad of adverse phenotypes, including lethality, increased cell death, cell cycle arrest, reduced fertility, organ development defects and tumorigenesis [93].

## 3. Roles of the Beak during Translation

The head of the small r-subunit is a flexible and dynamic structure involved in the engagement of the mRNAs and tRNAs during translation. Taking into consideration the strategic position of the beak at the entrance of the mRNA channel in the small r-subunit, it is not surprising that the beak has been linked to diverse functions during the translation process:

(i) During translation initiation, the beak is an important site for the interaction of *trans*-acting factors both in prokaryotic and eukaryotic ribosomes. For example, cryo-EM has revealed that the bacterial aldehyde-alcohol dehydrogenase E (AdhE) enzyme interacts with ribosomes in the beak region [94]. This enzyme provides a further RNA helicase activity, in addition to the intrinsic one of the ribosome [95], in order to ensure the linear configuration of structured mRNAs at the mRNA entrance to facilitate their translation [94]. In eukaryotes, several RNA helicases play roles during translation initiation. The canonical initiation factor eIF4A and the Ded1 (DDX3 in mammals) RNA helicase assist in the unwinding of the 5′-UTR secondary structure of most mRNAs [96,97,98]. Interestingly, in mammals, the translation initiation of endogenous and viral mRNAs with highly structured 5′-UTRs requires an additional RNA helicase, named DHX29 [99,100]. As revealed by cryo-EM, DHX29 contacts the beak and adjacent regions by interacting with at least uS3, eS10 and eS12 [101]. In other examples, the beak has been described as being important for the recognition of specific mRNAs. Accordingly, mammalian eS10 has been found to specifically interact with a class of cellular mRNAs containing the so-called TISU-element in their short 5′-UTRs [102].

(ii) The loading of the mRNA itself into the mRNA channel of the small r-subunit during translation initiation is regulated by the opening and closing of an mRNA latch situated below the beak that connects the body and the head of the small r-subunit [74]. The open conformation of this structure is promoted by the binding of distinct initiation factors (eIF1 and eIF1A in eukaryotes, IF1 in prokaryotes) to the small r-subunit during the formation of the eukaryotic 43S pre-initiation complex [103]. eIF1A is a globular protein harboring unstructured N- and C- terminal extensions of ca. 25 amino acids. During translation initiation, the globular domain of eIF1A is positioned at the A-site, while its extensions seem to project out of this site of the ribosome; thus, preventing tRNA binding to this site [103]. From X-ray crystallography, it can be inferred that the N-terminus of eIF1A directly contacts the eukaryote-specific N-terminal extension of eS31 and approaches extensions of other r-proteins, such as eS10, uS3 and uS19, that are adjacent to each other in the A-site [104]. These interactions seem to be crucial for translation as mutations in the N-terminal tail of eIF1A, which are frequently observed in several types of cancers [105], result in reduced binding of eIF1A to its r-protein partners and a hyperaccurate recognition of AUG codons that are embedded in an optimal sequence context [106,107]. Importantly, the phenotypic analysis of yeast *ubi3* mutations (e.g., *ubi3G75,76A*) that interfere with the cleavage of the ubiquitin-eS31 fusion protein indicates that the non-cleaved protein can still assemble into mature 40S r-subunits, which are active in translation but mildly defective at the translation initiation stage. This defect is likely due to the interference of the ubiquitin moiety with the binding and proper activity of the initiator tRNA and the eIF1A factor [89,104]. Moreover, interactions between several subunits of the eIF3 complex and beak components, including eS10 (e.g., [108]), have been described to occur within the yeast 43S pre-initiation complex; these are expected to be of functional relevance during translation initiation (e.g., [109]).

(iii) The beak also participates in the formation of the binding surface for the internal ribosome entry site (IRES) of some viral mRNAs on human 40S r-subunits. For instance, among other 40S r-proteins, eS10 contributes to the binding of the hepatitis C virus (HCV) IRES [110]. Viral proteins also interact with the beak to hijack the host’s translation machinery. One interesting example concerns the SARS-CoV-2 coronavirus, whose non-structural protein NSP1 contains a globular N-terminal domain that binds the base of the beak, while its C-terminal extension blocks the mRNA channel entry site and thereby prevents any mRNA accommodation; thus, inhibiting translation of host mRNAs [111,112,113]. However, NSP1 does not impede the translation of viral mRNAs, which is promoted by the presence of a *cis*-acting RNA hairpin in the 5′-UTR of these mRNAs [114,115].

(iv) Translation is reversibly shutdown upon nutrient starvation in a variety of ways, including the accumulation of inactive or hibernating vacant 80S ribosomes. These dormant ribosomes contain eEF2·GTP in the A-site and the hibernation factor Stm1 (in yeast) or SERBP1 (in mammals) in the mRNA channel, thereby impeding mRNA binding [87,116]. This mechanism of blocking the mRNA entry tunnel resembles that mentioned above for the coronavirus NSP1 protein. It has been described that the C-terminal region of Stm1/SERBP1 also stably associates with the head of 40S r-subunits, likely via binding to eS10, eS12 and eS31 [87,116].

(v) In all ribosomes, the interaction of the different translation elongation factors with the ribosome leads to specific movements of the head domain of the small r-subunit, including that of its associated beak towards the shoulder of the body of the same r-subunit (e.g., [117,118]). In eukaryotes, the beak components themselves interact with distinct domains of the translation elongation factors, as exemplified by the interaction of eS12 and eS31 with domains II and IV of eEF2 [7]. In agreement with the important role of beak r-proteins in the fidelity of translation elongation, the depletion of the essential yeast eS10 as well as the mutation or deletion of the genes encoding yeast eS12 and eS31 result in translation defects, including misreading [76,88,119]. Moreover, due to the specific position of the N-terminal extension of eS31 in the A-site of the ribosome, the assembly of non-cleaved yeast Ubi3 is expected to sterically interfere with the binding of the translation elongation factors to the ribosomal GTPase-associated center [89,91].

(vi) The beak is also expected to be functionally relevant during translation termination. Cryo-EM structures have revealed how eukaryotic translation termination factor 1 (eRF1), whose overall shape resembles a tRNA molecule, interacts with a stop codon in the A-site of the ribosome via its N-terminal lobe (e.g., [120] and references therein). Notably, in the structures of pre-termination complexes, a short segment of the N-terminal lobe of eRF1 is in close proximity to the initial residues of the N-terminal extension of eS31 [120,121]. Moreover, the mini-domain of eRF1, which is an insertion within the C-terminal domain, also interacts with the N-terminal extension of eS31 and protrudes toward the beak where it contacts helix h33 [120,121,122].

(vii) Another example of the role of the beak components in translation comes from studies of the cellular responses to elongation stalls induced by different stresses (including oxidative stress, heat shock or starvation) as well as by particular sequences, strong secondary structures and chemical damage within mRNAs. Normally, when exposed to stressful conditions, cells adapt by halting or decreasing the global synthesis of new proteins, while, concomitantly, inducing the selective translation of mRNAs encoding proteins that are necessary for cell survival and stress recovery [123]. This translational reprogramming can be mediated by multiple parallel and independent signaling pathways that converge on the modulation of the function of a few key translation factors [124]. Relatively recent studies from the Silva laboratory showed that following oxidative stress, induced by an exposure of yeast cells to hydrogen peroxide, a set of r-proteins were K63-specifically polyubiquitinated at different residues by the ubiquitin-conjugating enzyme Rad6 and the ubiquitin-protein ligase Bre1, with the extent of this modification declining very rapidly during stress recovery [125,126]. Most of these r-proteins are located within the head of the small r-subunit of the ribosome and include uS3, the beak components eS10, eS12 and eS31 and the P-stalk proteins uL10, uL11 and P2 [127]. Although oxidative stress induces a rapid inhibition of translation initiation via activation of the Gcn2 kinase (see below), K63-linked ubiquitination of r-proteins leads to an additional response, which results in the stalling of translation at the elongation stage [127]. Using cryo-EM and cryo-electron tomography, the Silva laboratory was also able to demonstrate that K63-linked ubiquitination of ribosomes alters the conformation of distinct r-proteins, including eS31 and eS12, that are located at the interface of the two r-subunits where eEF2 binds, thereby interfering with its efficient binding and/or GTPase activity and promoting the translational halt at the elongation stage, specifically at the rotated pre-translocation stage 2 [128].

If a ribosome persistently stalls on an mRNA, collisions with the trailing ribosomes will eventually occur; this phenomenon triggers different ribosome-associated quality control (RQC) mechanisms. It is thought that these mechanisms have evolved in order to relieve stalled ribosomes, thus avoiding the depletion of active ribosome and tRNA pools, which would prevent their participation in new rounds of protein synthesis and could reduce cellular fitness or survival [129,130]. RQC mechanisms additionally target damaged mRNAs and incomplete polypeptide chains for degradation [129]. In prokaryotes, the rescue of ribosomes stalled at the 3′ end of mRNAs lacking a stop codon, thus containing an empty A-site, often involves the action of the long transfer-messenger RNA (tmRNA), whose interaction with the ribosome occurs through the formation of a ring of its large loop around the beak of the 30S r-subunit (for further information, see [131]). In eukaryotes, different rescue pathways center around the recognition of the empty A-site in the ribosome (e.g., [130,132]). The rescue of eukaryotic ribosomes stalled on truncated and aberrant mRNAs lacking stop codons (NSD, non-stop decay) relies on several factors, such as Dom34 (Pelota in mammals), Hbs1 (HBS1L in mammals), Rli1 (ABCE1 in mammals) and Ski7 (HBS1L3 in mammals). These factors interact or have the potential to interact with ribosomes in a similar manner as translation elongation and termination factors [133]; therefore, it is expected that the binding and function of these factors, and, thus, the fate of NSD, could be altered by mutations affecting beak components. Dom34/Pelota is structurally related to tRNAs and eRF1 and binds the ribosome in a similar way to these two; in turn, Hbs1, which interacts with Dom34/Pelota, is a member of the family of translational GTPases that includes eEF1, which delivers aminoacyl-tRNAs to the A-site, eRF3, which interacts with eRF1 in a similar manner to Hbs1 with Dom34, and Ski7, which is a paralog of Hbs1 [133]. The N-terminal domain of Ski7 mediates the recruitment of the exosome and the Ski2-Ski3-Ski8 complex (SKIV2L-TTC37-WDR61 in humans) [134,135,136], while its C-terminal part contains the GTPase-like domain that it is assumed to interact, similar to other translation GTPases, with the GAC site of the ribosome, but whose exact role is still unknown [137,138]. Cryo-EM structures of different ribosomes, Ski2/3/8 complex and exosome intermediates suggest a scenario where a stalled ribosome bound to the Ski2/3/8 complex recruits a pre-assembled exosome-Ski7 complex [136,139,140]. Through this triple (ribosome–Ski2/3/8 complex–exosome) interaction, aberrant mRNA substrates are unwound and guided into the exosome. The ribosome-bound Ski2/3/8 complex specifically recognizes the 40S r-subunit by binding near the entry of the mRNA channel and connecting the head and beak regions [140]. Concerning the beak, the Ski2 helicase interacts with several r-proteins, among them eS10 and uS3, while the N-terminal part of Ski3 contacts eS12 [139,140].

Prolonged ribosome stalling leads to a ribosome collision of the trailing ribosomes with the stalled ribosome [141,142]. Under these circumstances, the E3 ligase Hel2 (ZNF598 in mammals) recognizes the collided ribosomes and adds ubiquitin to a number of 40S r-proteins at precise lysine residues, among them eS10 and uS3 ([143] and references therein). This ubiquitination is assumed to serve as the starting signal for the progression of the RQC response in order to dissociate the stalled ribosomes into r-subunits and degrade their associated mRNAs and nascent peptides [144]. In addition, beside many other responses [142], ribosome collisions also activate the kinase Gcn2, and evidence suggests that this activation can occur independently of the presence of deacetylated tRNAs [145,146], which constitutes its classical activation pathway. This activation is dependent on Gcn1, and cryo-EM has nicely revealed how this long, tube-like HEAT repeat protein spans across a collided disome by forming an extensive network of interactions both with the leading and trailing ribosome [147]. Interestingly, along its interaction path, the region preceding the central eEF3-like HEAT repeats engages in contacts with eS10, eS12 and eS31 within the beak of the 40S r-subunit of the colliding ribosome [147,148]. Perturbations of the beak, elicited by absent or mutated beak r-proteins, could influence Gcn1 such that its ribosome association or Gcn2-binding capacity, both of which are required for Gcn2 activation, is affected. In turn, activated Gcn2 can then phosphorylate eIF2α to downregulate general translation initiation and to enable the translation of specific mRNAs, such as those encoding Gcn4 in yeast or ATF4 in mammals, in order to adequately respond to the stress that causes ribosomes to collide. In line with the structural integrity of the beak being necessary for an efficient Gcn1-mediated Gcn2 activation, lower levels of eS10 (individual deletion of *RPS10A* or *RPS10B*) or the absence of eS31 were shown to reduce the extent of eIF2α phosphorylation or to impair derepression of *GCN4* mRNA translation in response to amino acid starvation, respectively (e.g., [148,149]). Recently, the ubiquitination of eS31 has also been reported to occur in circumstances of translation elongation inhibition where the A-site is occluded by a trapped eEF1A factor bound to an aminoacyl-tRNA [150]. In this case, the reaction is dependent on an E3 ligase called RNF25. Ubiquitination of eS31 is required for the degradation of the trapped eEF1A, which itself is ubiquitinated both by RNF25 and an additional E3 ligase RNF14, with the latter directly interacting with GCN1, which is also essential for eEF1A degradation [150].

## 4. Other Cellular Functions of the Beak Components

The beak r-proteins, in addition to their clear role in translation, participate in other cellular processes, including ribosome biogenesis (see Section 5), activation of the p53-dependent pathway in response to nucleolar stress as well as oncogenesis (see Section 6), and cell competition (see below). Whether or not the effects that mutations in the beak r-proteins have on these processes are translation-dependent or independent is still unclear in some cases.

As mentioned above, a systematic study has analyzed the contribution of loss-of-function mutations for most r-proteins, including eS10, eS12 and eS31, to multiple phenotypic features in six relevant eukaryotic model organisms (*S. cerevisiae*, *C. elegans*, *D. melanogaster*, *D. rerio*, *M. musculus*, and *H. sapiens*) [93]. Several reports on the characterization of specific features of eS10 have highlighted the important role of this r-protein. These include the description of a hypo-proliferative phenotype, known as the Minute phenotype, associated with loss-of-function mutations in one of the two copies of the *RPS10* gene during development in *Drosophila*. The Minute phenotype is characterized by a prolongation of the developmental time, the presence of short and thin bristles, and reduced fertility [151]. In *Drosophila*, eS10 is encoded by duplicated genes, and, interestingly, the expression of one of the two genes is enriched in the germline cells of embryonic gonads, suggesting a germline-specific role [151]. In *Arabidopsis*, loss-of-function mutations in one of the genes encoding eS10 lead to a reduction in stamen number, shoot and floral meristem defects, and a leaf polarity deficiency [152].

The r-protein eS12 also plays interesting roles not directly related to ribosome biogenesis or translation. In specific neurons, *RPS12* mRNA levels, among other r-protein transcripts, seem to be reduced by an acute period of sleep deprivation (hippocampus) or injury of the sciatic nerve (dorsal root ganglion), suggesting dynamic changes in ribosome composition following these insults (reviewed in [153]). In *S. cerevisiae*, a specific mutation in the *RPS12* gene leads to the suppression of phenotypes elicited by rDNA instability upon Fob1 overexpression [154]. Whether this phenomenon is the result of an extra-ribosomal function of yeast eS12 remains to be explored. Undoubtedly, the most interesting function of eS12 besides its orthodox roles in ribosome biogenesis and translation is that related to cell competition in *D. melanogaster*. In the classical form of cell competition, wild-type cells (homozygotic cells for r-protein genes; hereafter Rp^+/+^ cells) in genetic mosaic flies are able to actively eliminate their adjacent heterozygotic cells (heterozygotic cells for r-protein genes; hereafter Rp^+/−^ cells) from imaginal discs via apoptosis [155,156,157]. In this process, eS12 plays a specific role as a sensor of an imbalance of r-proteins to allow the elimination of Rp^+/−^ cells [158]. Thus, the viable missense *rps12*[G97D] mutant allele of *RPS12* in homozygosis prevents cell competition of Rp^+/−^ cells by wild-type Rp^+/+^ cells [158,159]. In other words, *rps12*[G97D]^+/+^ Rp^+/−^ cells are not eliminated by wild-type Rp^+/+^ cells. Moreover, the relative copy number of the wild-type *RPS12* allele in Rp^+/−^ cells is apparently what determines the competitiveness [158]. It has been shown that the eS12[G97D] variant efficiently assembles into 40S r-subunits [158]. Moreover, the yeast *rps12*[G102D] allele (equivalent to *Drosophila rps12*[G97D] allele), when it is the sole cellular source of eS12 r-protein, neither confers a growth defect nor a global impairment of translation (S. M.-V., unpublished results). Interestingly, it has been shown that *Drosophila* eS12 is required to increase the transcription of the gene encoding the transcription factor Xrp1 [159,160], which itself also directly regulates cell competition [156,159]. Consistently, loss-of-function mutations in Xrp1 also prevent competition of Rp^+/−^ by wild-type Rp^+/+^ cells [159]. Whether the function of eS12 in promoting Xrp1 expression is extra-ribosomal still needs confirmation.

As for eS10 and eS12, there are reports indicating that eS31 could also have ribosome-independent functions (for a recent review, see [161]). First, eS31 has been identified as a regulator of the LMP1 protein encoded by the Epstein-Barr virus (EBV); thus, eS31 binds directly to LMP1 and increases its stability by reducing its proteasome-mediated degradation [162]. Moreover, overexpression of eS31 leads to increased cell growth and survival as the result of LMP1-mediated oncogenic events (e.g., epithelial to mesenchymal transition, motility, migration and invasion). In addition, as *RPS31* (also known as *RPS27A)* mRNA levels are reduced in sperm with low motility, eS31 might be necessary for optimal sperm functionality in humans [163]. In plants, eS31 seems to be highly expressed in meristematic tissues, pollen and ovules [164], and flowers of *RPS31*-silenced plants exhibit abnormal development [165]. Finally, it has been observed that double-strand break DNA damage results in the MDM2-independent proteasomal degradation of eS31 in HEK293 human cells, and as a consequence, these cells contain ribosomes that specifically lack eS31 and exhibit lower global translation activity [166]. Whether this phenomenon is part of an adaptive response to deal with DNA damage remains to be determined [166].

## 5. Assembly and Maturation of the Beak Structure

The assembly of the beak has been analyzed in the context of the general maturation of the 40S r-subunit, which begins in the nucleolus and ends in the cytoplasm, both in yeast and in human cell lines (for a review, see [167]). Moreover, given the fact that the beak is a pronounced protrusion in the structure of the 40S r-subunit, the assembly of the beak represents a challenge for the nucleocytoplasmic transport of this r-subunit. Using genetics in yeast and siRNA technology in human cell lines, the role of the beak r-proteins in pre-rRNA processing and r-subunit assembly has also been well examined. In both cases, it has been described that the three r-proteins that form the eukaryotic beak are required for the production and the stability of mature 40S r-subunits.

In yeast, eS10 is an essential r-protein whose contribution to ribosome biogenesis has been assessed by the use of a yeast strain conditionally expressing this r-protein [86,168]. These studies showed that eS10 is required for the efficient maturation of the 20S pre-rRNA, which accumulates to high levels upon eS10 depletion both within nucleoplasmic pre-40S r-particles, as a consequence of a delay in the export of pre-40S r-particles and cytoplasmic pre-40S r-particles, and as a consequence of inefficient 20S pre-rRNA processing at site D [86,168]. Interestingly, knocking down the expression of human *RPS10* leads to cytoplasmic accumulation of the 18S-E pre-rRNA, which is the equivalent human form of the yeast 20S pre-rRNA [169,170]. In yeast, we and others have demonstrated that the quasi-essential r-proteins eS12 and eS31 are also crucial for the efficient cytoplasmic processing of the 20S pre-rRNA into mature 18S rRNA [88,89,92]. Similarly, the cytoplasmic accumulation of 18S-E pre-rRNA has also been reported upon siRNA-mediated knockdown of human *RPS12* or *RPS27A* expression [170,171,172].

The precise timing of the assembly of the beak rRNA and r-proteins has also been analyzed in both yeast and humans at a reasonable resolution. Assembly of this structure involves compositional and structural changes of both the beak rRNA and r-proteins. In general terms, while the timing of eS31 assembly (nucle(ol)ar or cytoplasmic) is still controversial, it is likely that eS12 is incorporated early during the formation of 90S pre-ribosomal particles and that eS10 assembly occurs within late cytoplasmic pre-40S r-particles (see below). In yeast and the fungus *Chaetomium thermophilum*, the structural analysis of 90S pre-ribosomal particles suggests that the four subdomains of the 18S rRNA (5′, central, 3′ major, and 3′ minor) fold independently and associate co-transcriptionally with a set of r-proteins and ribosome assembly factors (RAFs), before being compacted into a defined pre-ribosomal particle. Analysis of the first reported structures of 90S pre-ribosomal particles indicated that the folding of the 3′ major domain of the 18S rRNA, the helix h33 included, requires the prior co-transcriptional structuring of the 5′ domain [173,174,175]. However, a more recent structural determination of a series of 90S assembly intermediates from *C. thermophilum* provides evidence that the formation of the 90S does not follow a strict 5′ to 3′ co-transcriptional direction; instead, the 3′ major and 3′ minor domains seem to assemble first with the 5′-ETS domain of 35S pre-rRNA, preceding the incorporation of the 5′ and central domains of pre-18S rRNA into 90S pre-ribosomal particles [176]. In any case, from the diverse collection of structurally stable 90S r-particles available in the literature, it is clear that these particles contain a clearly identifiable, immature beak structure. In most of these particles, the nascent beak structure comprises eS12, while only a few of them also contain eS31. As an example of this, in the 90S structure reported by Sun et al. [173], the beak forms a protrusion and it is composed of helices h32-34 and the r-proteins eS12 and eS31, connected to the body of the particle by the RAF Emg1 (see Figure 5). At this level, it is also clear that the presence of Enp1, which stabilizes the beak by binding to helices h32-34, impedes the incorporation of eS10, which can only occur after the release of Enp1 from late pre-40S r-particles in the cytoplasm [177,178]. Moreover, the fact that eS31 (and to a lesser extent eS12) is not present in many of the structural maps of 90S r-particles available in the literature, as well as in a variety of further pre-40S r-particles (see below), clearly indicates that the association of these r-proteins with early precursors of 40S r-subunits might be highly labile and should only become stable during late and cytoplasmic steps of 40S r-subunit maturation, concomitant with the formation of a more rigid beak structure. In this regard, another RAF, Tsr1, apparently blocks the correct binding of eS31 until its repositioning at a late maturation step occurring on cytoplasmic pre-40S r-particles [179]. Alternatively, as favored by other authors, eS12 and especially eS31 are only incorporated into cytoplasmic pre-40S r-particles [179,180]. However, at least in the case of eS31, we have identified a functional nuclear localization signal (NLS) within the first 25 amino acids of the N-terminal extension of yeast eS31, which is conserved in other eukaryotes [76]. This sequence is sufficient to target a triple GFP reporter to the nucleus, and most importantly, a functional GFP-tagged eS31 protein notably accumulates in the nucleus upon depletion of different 90S RAFs, among them Emg1, which leads to the nuclear retention of pre-40S r-particles ([76] and S. M.-V., unpublished results). Whatever the case may be, the N-terminal ubiquitin moiety present in the linear precursor of eS31 in many eukaryotes, including yeast and humans, is very rapidly and efficiently processed. Accordingly, under wild-type conditions, the Ubi3 precursor has so far never been detected; hence, it must be processed prior to the incorporation of eS31 into pre-40S r-subunits [89,92,181]. Consequently, it is unlikely that the ubiquitin moiety fused to eS31 directly participates in the ribosomal assembly of eS31. Moreover, when a wild-type and a cleavage-deficient Ubi3 variant are co-expressed in the same cells, eS31 derived from wild-type Ubi3 is preferentially incorporated into pre-40S r-particles compared to the non-cleaved ubiquitin-eS31 fusion protein, which in turn is rapidly degraded [89]. Forcing the assembly of non-cleaved Ubi3 into nascent 40S r-subunits only mildly impairs their biogenesis, but, as mentioned above, may lead to translation initiation defects [89].

Following the sequential cleavages at sites A_0_–A_2_ within the 35S pre-rRNA, the yeast 90S r-particle is dismantled and converted into an early nuclear pre-40S r-particle, which is rapidly exported to the cytoplasm. At this step, and before export, most 90S RAFs have disassembled and only a few others have been recruited, among them Rio2, Tsr1, Ltv1 and Rrp12 [182]. Perhaps the characteristic that defines best the nucleoplasmic pre-40S intermediates is the high flexibility of their head domain, which becomes more structured as the particles transition through their maturation [183,184,185]. The incorporation of Ltv1 is relevant for beak formation as it interacts with Enp1 and the r-protein uS3, which binds at the base of the beak structure [119,186,187]. The recruitment of uS3 is initiated just before or concomitant with that of Ltv1 [188,189]. The r-protein uS3 consists of two distinct N- and C-terminal domains and is delivered to nuclear pre-40S r-particles by its dedicated chaperone Yar1 [190]. Yar1 binds only the N-terminal domain of uS3; thus, initial interaction of uS3 with pre-40S r-particles likely occurs through its C-terminal domain [188,191,192]. The release of Yar1 is concomitant with the interaction of the N-terminal domain of delivered uS3 with Ltv1. This interaction also contributes to preventing uS3 from prematurely acquiring its final and stable position within cytoplasmic pre-40S r-particles, which is only achieved upon the global structural changes occurring in these r-particles after Ltv1 release [119,179,190,191]. Indeed, a subcomplex formed by Ltv1, Enp1 and uS3 can be untethered from purified yeast pre-40S r-particles at high salt concentrations, while uS3 cannot be extracted from mature 40S r-subunits by the same treatment [193], indicating that uS3 is less stably integrated into pre-40S than mature 40S r-subunits. It has been suggested that a certain degree of flexibility in the beak is required at this nucleoplasmic stage because a rigid beak structure close to the head of the pre-40S r-subunit might hinder export through the nuclear pore complex (NPC) [179,193,194,195].

Once exported, the early cytoplasmic pre-40S r-particles undergo a cascade of maturation events, the first ones being essential for beak formation. The precise chronology of these events remains to be elucidated at high resolution, but it seems that it first involves the recruitment of the casein kinase Hrr25, also favored by the previous binding of the uS3 r-protein [190,192]. Moreover, the direct interaction between Hrr25 and Ltv1 appears to weaken the association between Ltv1 and Enp1 [192]. Hrr25, which is an essential protein, then phosphorylates Ltv1 on specific conserved serine residues, leading to the release of Ltv1 from pre-40S r-particles [188,189,192]. Strikingly, Hrr25 is no longer essential in the absence of Ltv1 or upon phosphomimetic substitutions of the specific Ltv1 serine residues [188,189], indicating that the essential function of Hrr25 is linked to Ltv1 in ribosome biogenesis. Interestingly, the release of Ltv1 is coordinated with that of Rio2 on the intersubunit side of the head domain of the pre-40S r-particle; a process that is mediated by the correct assembly of the uS10 r-protein [192]. The release of Ltv1 now provokes the dissociation of Enp1, which is also phosphorylated by the Hrr25 orthologue (CK1δ/ε) in humans [196]; phosphorylation of yeast Enp1 by Hrr25 is still controversial [189,193]. The dissociation of Enp1 and Ltv1 is absolutely required for nascent 40S r-subunits to become translationally competent, as their interaction with the beak environment would hinder the opening of the mRNA channel [177]. Another consequence of the dissociation of these factors is that eS10 gains access to its binding position and is integrated into the beak structure [180,186,197]. Concomitantly, uS19 and the two domains of uS3 are fitted into their mature position [119,179,192]. All these events promote the structural organization of the beak, which then enables the progression of the maturation events in other regions of the pre-40S r-particles [167,180].

Formation of the beak in human 40S r-subunits seems to occur in a similar way to that described in yeast, albeit with certain peculiarities [167,197,198]. Orthologs for all key factors mentioned above have been described in humans, including Enp1, Ltv1 and Hrr25 (Bystin, LTV1 and CK1δ/ε, respectively) [199]. Most importantly, despite differences in pre-rRNA processing between yeast and humans [200], the positioning, timing of interaction and dissociation, and function of all these RAFs have been well conserved [199]. Moreover, the structures of several nuclear and cytoplasmic pre-40S r-particles have been described, providing detailed insights into the maturation steps [184,197,198,201]. As an example, the cryo-EM structures of apparently nucleoplasmic human pre-40S r-particles contain, in addition to the orthologs of Enp1 and Ltv1, the r-proteins eS12, eS31 and uS3, despite the fact that, as in yeast, the assembly time point of eS31 and eS12 is again controversial [184,197].

Finally, although beyond of the scope of this review, the assembly of the prokaryotic beak is also an important late event during the maturation of 30S r-subunits; in this regard, it is interesting to mention that some authors have proposed that the bacterial assembly factor RimM, which is involved in the maturation of the 3′ domain of the head of 30S r-subunits (see [202] and references therein), works as a functional analog of Ltv1 in bacteria [119,189]. Strikingly, in the absence of RimM, the 30S beak (h33) is not correctly folded, and several r-proteins, including uS19 as well as the tertiary binders uS3 and uS10, do not efficiently assemble into pre-30S r-particles [203,204]. Moreover, the assembly of the small subunit of the trypanosomal mitoribosome represents a major challenge for those researchers who are studying this process (e.g., [205]).

## 6. Beak Components and Human Diseases

It is evident that mutations and dysregulation of the majority of r-protein genes are linked to a range of human genetic diseases, such as ribosomopathies and cancer [206,207]; in this regard, beak r-proteins are not an exception.

Ribosomopathies are a group of rare inherited or acquired genetic diseases linked to defects in r-proteins or ribosome biogenesis factors [207,208,209]. Despite the importance of ribosomes in all cell types, these diseases result mainly in tissue-specific manifestations, especially in the hematopoietic system [207]. Intriguingly, inherited ribosomopathies are congenital and normally exhibit a paradoxical transition from early symptoms related to cellular hypo-proliferation to a hyper-proliferative oncogenic state later in life [210]. The best studied ribosomopathy, which is also one of the most prevalent ones (10 individuals per million live births) is the so-called Diamond–Blackfan Anemia (DBA) [211]. DBA is mostly a dominant genetic disorder (autosomal or X-linked) that is characterized by the reduced formation of red blood cells and is also associated with a series of other congenital anomalies, such as skeletal abnormalities, heart and genitourinary malformations, and an increased cancer susceptibility [212]. Most patients diagnosed with DBA harbor heterozygous loss-of-function mutations in particular genes encoding r-proteins, either of the small or the large r-subunit [212]. About 3% of all DBA patients have been reported to carry mutations in the *RPS10* gene, most likely leading in all cases to the production of non-functional eS10 variant proteins [212,213]; these mutations mostly consist of (i) insertion mutations that cause a frameshift and the appearance of a premature termination codon, (ii) nonsense mutations, namely, changes of particular codons to a stop codon (often, the R113Stop mutation), (iii) missense mutations that transform the *RPS10* start codon into an isoleucine or a threonine codon (M1I or M1T), and (iv) different other missense mutations (e.g., L14F, P30L) of so far unknown biological significance ([213,214]; for more details, check the UniProt entry P46783 and the OMIM entry 603632). Mutations in the human *RPS27A* gene, which codes for human eS31, have also been identified in patients with DBA (e.g., S57P); however, whether these mutations are indeed pathogenic genetic variants remains to be determined [215]. To our knowledge, no DBA-linked mutations have so far been reported in the *RPS12* gene. However, as the underlying mutations in at least 20% of patients with DBA syndromes have not yet been identified [212], it is still possible that *RPS12* alleles could be responsible for DBA manifestations; in line with this possibility, *RPS12* haploinsufficiency in mice leads to an erythropoiesis defect that recapitulates the one found in DBA patients (discussed in [216]). From a very simplistic point of view, as the DBA disease is mostly caused by loss-of-function mutations in several r-protein and a few RAF genes, all associated DBA symptoms and manifestations must come from common dysfunctions of the same molecular process that, in this case, can be no other than ribosome biogenesis [217], ultimately leading to an impairment or a limitation of translation. Thus, in a non-exclusive manner, it has been proposed that (i) the hypo-proliferative, pro-apoptotic anemia associated with DBA could be the consequence of a global reduction in translation, limiting below a critical threshold the synthesis of critical proteins, such as the globins and the transcription factor GATA1, with the latter being essential for normal erythropoiesis [218]. In agreement with this possibility, *GATA1* translation is reduced in erythroid precursor cells of DBA patients with mutations in different r-protein genes (e.g., [219,220,221]), and loss-of-function mutations in *GATA1* result in a DBA-like phenotype (e.g., [222,223] and references therein). (ii) DBA cells display elevated levels of reactive oxygen species (ROS), which, by generating a high oxidative stress, inhibit cell proliferation [206,207]. In this regard, lowering cellular ROS levels by antioxidants can rescue the proliferation defects in cells subjected to r-protein haploinsufficiency or carrying selected r-protein mutations [207,224]. (iii) As a consequence of the ribosome biogenesis deficiency occurring in DBA cells, the so-called nucleolar stress response is triggered in these cells, which induces p53 stabilization and enables p53 to transactivate its target genes, leading to cell cycle arrest, apoptosis, autophagy, and senescence [225,226]. In normal growth conditions, cellular levels of p53 are maintained low due to its efficient recognition and ubiquitination by the E3 ubiquitin ligase MDM2 (HDM2 in humans) and the subsequent degradation of ubiquitinated p53 by the proteasome. However, when ribosome biogenesis is impaired, non-assembled r-proteins tend to accumulate and can be released from the nucleolus to the nucleoplasm. Several different free r-proteins, but primarily uL5 and uL18 as part of the 5S RNP, can bind and sequester MDM2, thereby preventing the degradation of p53; thus, the upregulation of p53 explains many of the hypo-proliferative phenotypes displayed by DBA patients, including bone marrow erythroid hypoplasia [206,227,228]. In consonance with the relevant role of nucleolar stress in DBA, genetic or pharmacological inactivation of p53 can rescue disease-associated phenotypes [206,209,225,229]. Interestingly, it has been shown that eS31 is able to regulate the MDM2-p53 loop in response to nucleolar stress [230,231]. Moreover, eS31 apparently interacts with the central acidic domain of MDM2 through its eukaryote-specific N-terminal extension [230]. Therefore, this interaction seems not to be mutually exclusive from the ones of MDM2 with p53, which used a short, N-terminal segment to bind to the N-terminal domain of MDM2 [232], and with the r-proteins uL5 and uL18, which mostly involve the Zn-finger and RING domains of MDM2 [59,233]. Importantly, the overexpression of eS31 reduces MDM2-mediated ubiquitination of p53, thereby leading to its stabilization and activation [230,234]. The induction of p53 by eS31 may likely be additionally fueled by the observation that the *RPS27A* gene is apparently also transcriptionally activated by p53 [234]. Moreover, it has also been shown that MDM2 mediates the ubiquitination and proteasomal degradation of eS31 in response to nucleolar stress, indicating that free eS31 could be a physiological substrate of MDM2 [230]. It has been proposed that this mutual inhibitory regulation between MDM2 and eS31 may contribute to cellular recovery after the experienced stress [230]. Another report has suggested that the RAF PICT1 seems to regulate the interaction between eS31 and MDM2, as low levels of PICT1 are apparently required for the efficient translocation of eS31 from the nucleolus to the nucleoplasm, so that it can bind MDM2 [235]. It has also been described in several human lung cancer cell lines that eS31 could interact with uL5 in a way that might weaken the strength of the interaction between uL5 and MDM2; thus, knockdown of *RPS27A* stabilizes p53 in a uL5-dependent manner, promoting the p53 tumor suppressor functions [231]. Altogether, the above-mentioned data highlight that eS31 is connected to the MDM2-p53 axis and suggest that eS31 may be relevant for fine-tuning the cellular response to and recovery from nucleolar stress. However, its importance is apparently cell-type dependent, as recently shown by the Schneider group, who demonstrated that knockdown of *RPS27A* robustly induced p53 in certain cell lines but not in others [181].

Cancer cells require a high production of ribosomes to sustain boundless growth and cell division (e.g., [236]). Moreover, many r-proteins, including the beak ones, have been implicated in cancer development (e.g., [206,237,238,239]). Mutations and altered expression of beak r-proteins have been described in many cancer types, in some cases likely displaying an extra-ribosomal function: (i) high expression of *RPS10* has been found in colorectal, renal and prostate cancer [240], whereas it has been reported that ribosomes purified from MDA-MB-231 breast cancer cells contain substoichiometric levels of eS10, among other r-proteins [119]. (ii) Overexpression of *RPS12* has been observed in colon adenomatous polyps and carcinomas as well as in gastric cancers [241]. Deletions of *RPS12* are frequently observed in diffuse large B cell lymphomas [242], and eS12 has been reported to play a role as a stimulator of WNT secretion in cancer cells, which is particularly important in the context of triple-negative breast cancer initiation and progression [243]. (iii) It has been reported that eS31 is overexpressed in renal, colon, cervical, and breast cancers, chronic myeloid leukemia and lung adenocarcinoma [161,231,244,245]. In most of these cases, the overexpression of eS31 correlates with poor prognosis for the patients. A recent review has highlighted the expression and role of eS31 in cancer cells and tumor tissues [161].

## 7. Concluding Remarks and Future Perspectives

Herein, we have discussed the relevance of the beak, a structurally conserved region of the small r-subunit of cytoplasmic ribosomes in all three domains of life. Notably, the beak has transitioned from a structure composed exclusively of rRNA in bacteria to a protuberance comprising three specific r-proteins, eS10, eS12 and eS31, in eukaryotes, while an intermediate situation prevails in archaea as the beaks of many species only contain two r-proteins, eS31 and eL8, with the latter being clearly the ancestor of the eukaryotic eS12 r-protein. As also discussed, the beak has diverged considerably in the mitoribosomes of some organisms, such as in trypanosomatids, perhaps owing to the particular translation requirements inside these organelles [246,247]. As outlined in this review, the beak has important roles in the three major phases of translation (initiation, elongation and termination) and is involved in many other events of the translation process, including ribosome stalling and collisions, as well as other seemingly translation-unrelated processes, such as cell competition in *Drosophila*. In many of these processes, the biological significance of post-translational modifications, such as ubiquitination, is still poorly understood. We have also highlighted the implication of the beak r-proteins in the stepwise assembly of nascent 40S r-subunits. Regarding the maturation of the eukaryotic beak, it is evident that further research is required to precisely define the assembly timing of the beak r-proteins; for instance, it is still controversial whether eS31 associates with 40S r-subunit maturation intermediates in the nucleus or the cytoplasm. More studies are also required to elucidate whether the beak r-proteins play active or passive roles during the assembly and nuclear export of pre-40S r-subunits. Finally, we discussed the relevance of the beak r-proteins in human diseases, especially ribosomopathies and cancer. A deeper understanding of the connection between these diseases and the ribosome biogenesis process is expected to offer new perspectives for therapeutic approaches.

## Figures and Tables

**Figure 1 biomolecules-14-00882-f001:**
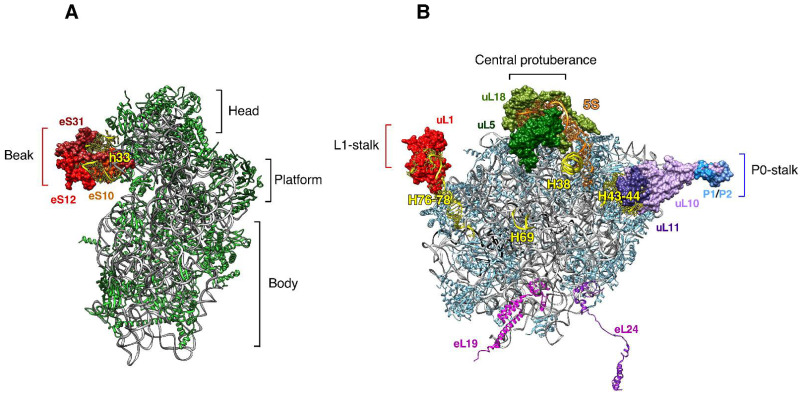
Protuberances of the r-subunits. (**A**) Structure of the mature 40S r-subunit of *Saccharomyces cerevisiae* showing its different structural domains (head, body, platform and beak) and highlighting the beak elements that include helix h33 of the 18S rRNA and the r-proteins eS10, eS12 and eS31. The 40S r-subunit is shown from the interface view; 18S rRNA is colored in gray (except h33 in yellow) and r-proteins not belonging to the beak structure are colored in green. Protein Data Bank (PDB) code: 4V88. (**B**) Structure of the mature 60S r-subunit of *S. cerevisiae* showing its different substructures (L1 and P0 stalks, and central protuberance) and highlighting the different protuberances: L1-stalk formed by helices H76-78 of 25S rRNA and the r-protein uL1; central protuberance formed by 5S rRNA and r-proteins uL5 and uL18; P0-stalk formed by helices H43-44 of 25S rRNA and the r-proteins uL11, uL10 (P0), P1 and P2 (two copies each). As other examples of protrusions found in the 60S r-subunits, we highlight helices H38 and H69 (yellow) and the long extensions of the r-proteins eL19 and eL24 (purple). The 60S r-subunit is shown from the interface view; 25S and 5.8S rRNAs are colored in gray and the rest of the r-proteins are colored in light blue. PDB code: 6OIG. Cartoons were generated using UCSF Chimera (https://www.rbvi.ucsf.edu/chimera; accessed on 1 June 2024).

**Figure 2 biomolecules-14-00882-f002:**
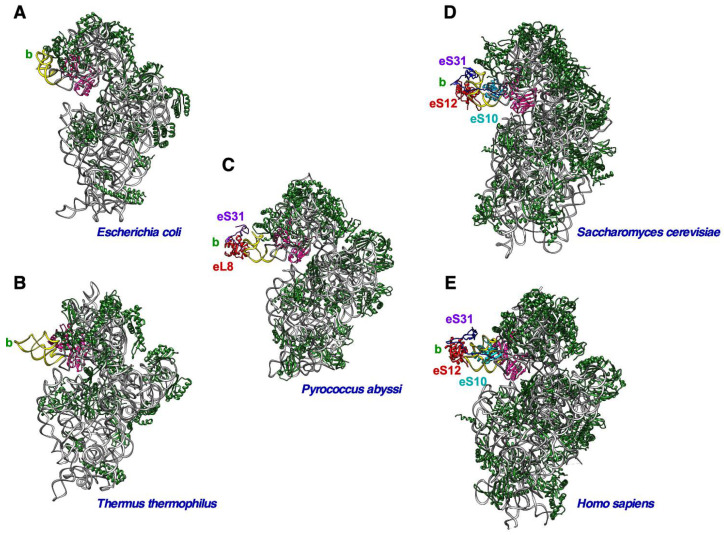
Comparison of small r-subunits of bacteria, archaea and eukaryotes. (**A**) Small r-subunit of *Escherichia coli*; PDB code: 7OE1; (**B**) small r-subunit of *Thermus thermophilus*; PDB code: 1J5E; (**C**) small r-subunit of *Pyrococcus abyssi;* PDB code: 7ZHG; (**D**) small r-subunit of *S. cerevisiae*; PDB code: 4V88; (**E**) small r-subunit of *Homo sapiens*; PDB code: 7R4X. In all cases, the interface view of the individual r-subunits is shown. The rRNA is colored in gray and the r-proteins in green. The beak (b) of all r-subunits is highlighted; helix h33 of rRNA is colored in yellow, and the r-proteins eS31, eL8, eS10 and eS12 in the indicated colors. The r-protein uS3, which is located at the base of the beak, is colored in pink. Cartoons were generated using UCSF Chimera (https://www.rbvi.ucsf.edu/chimera; accessed on 1 June 2024).

**Figure 3 biomolecules-14-00882-f003:**
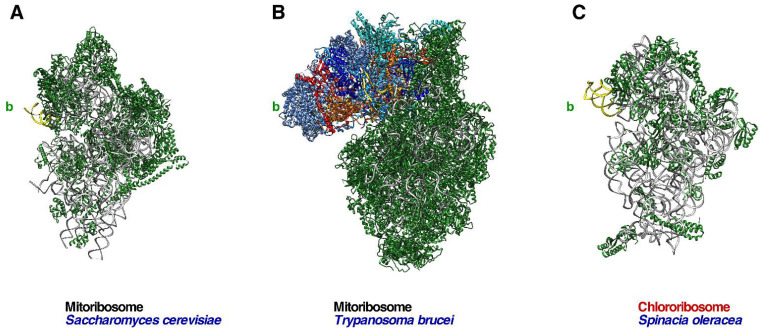
Comparison of the structural features of the small subunit of the mitochondrial ribosome from *S. cerevisiae* (**A**) and *Trypanosoma brucei* (**B**), and of the chloroplast ribosome from *Spinacia oleracea* (**C**). The PDB codes are 5MRC, 6HIW, and 5MMJ, respectively. In all cases, the interface view of the individual r-subunits is shown. The rRNA is colored in gray and the r-proteins in green. The beak (b) of all r-subunits is highlighted; helix h33 of rRNA is colored in yellow, and the set of specific r-proteins present in the *T. brucei* small r-subunit are shown in different colors. Cartoons were generated using UCSF Chimera (https://www.rbvi.ucsf.edu/chimera; accessed on 1 June 2024).

**Figure 4 biomolecules-14-00882-f004:**
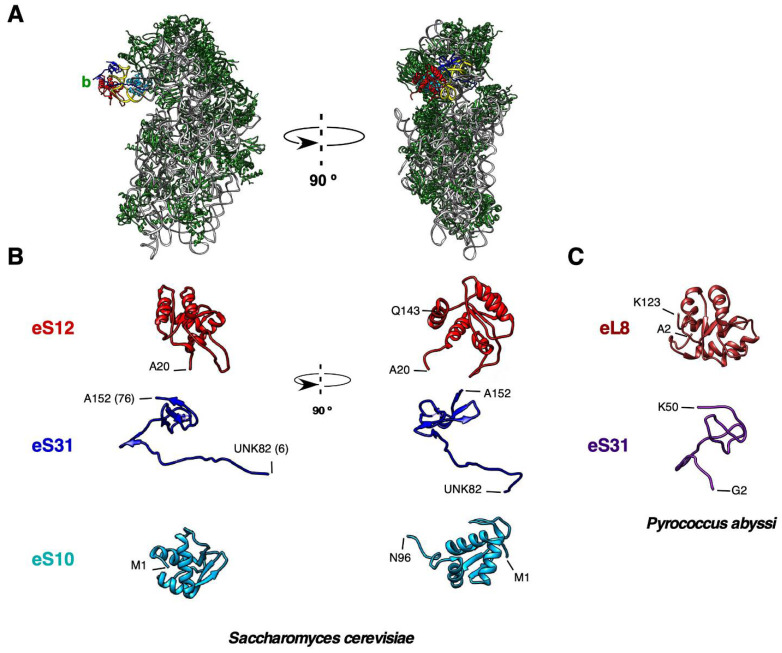
The r-proteins from the beak (b) of the yeast 40S r-subunit and the equivalent ones from the beak of the *Pyrococcus abyssi* 30S r-subunit. (**A**) Structure of the yeast 40S r-subunit (PDB code 4V88). (**B**) Structure of yeast eS12, eS31 and eS10 r-proteins are shown as found in the 40S r-subunit before or after 90° rotation on the Y-axis. (**C**) Structure of the equivalent r-proteins eL8 and eS31 from *P. abyssi* are also shown as found in the 30S r-subunit (PDB code 7ZHG). Note that the first 19 N-terminal residues of yeast eS12, the six first residues of yeast eS31, the last nine C-terminal residues of yeast eS10, the first residue of *P. abyssi* eL8 and the first and the last residue of *P. abyssi* eS31 are not present in the structures. Cartoons were generated using UCSF Chimera (https://www.rbvi.ucsf.edu/chimera; accessed on 1 June 2024).

**Figure 5 biomolecules-14-00882-f005:**
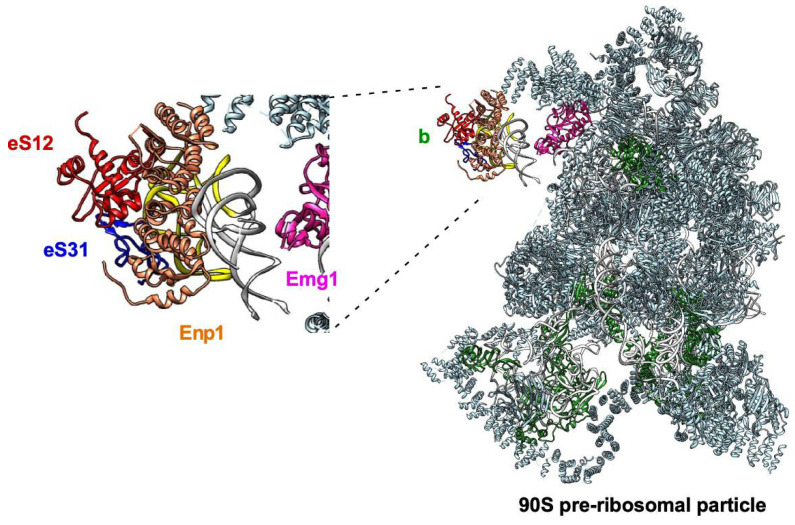
Formation of the early beak structure. Structural model of the yeast 90S pre-ribosomal particle (PDB code 5WYJ). The pre-rRNA is colored in gray, all r-proteins except eS12 and eS31 are colored in green and all ribosome assembly factors in light blue. Helix h33 is highlighted in yellow, eS12 in red and eS31 in blue. The interaction of Enp1 with the beak is shown. The beak is connected to the rest of the 90S particle by its interaction with Emg1/Nep1 (colored in pink). Left, close-up view of the beak (b) region. Cartoons were generated using UCSF Chimera (https://www.rbvi.ucsf.edu/chimera; accessed on 1 June 2024).

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
