# Peer review of "The Beak of Eukaryotic Ribosomes: Life, Work and Miracles"

_biomolecules, 2024, doi:10.3390/biom14070882_

Round 1

Reviewer 1 Report

Comments and Suggestions for Authors

The beak of the 40S ribosome is a crucial feature of the small subunit maturation process and plays a significant role in all aspects of translation. This review provides an overview of the protuberances of ribosome subunits, with a particular focus on the beak of the eukaryotic 40S subunit. The authors summarize recent advancements in the structure, function, and disease relevance of the 40S beak. Overall, the review is well-written, organized, comprehensive, and insightful.

However, there are a few improvements that can be made. Firstly, when discussing the composition of the beak, the authors used many recent structures obtained from X-ray crystallography or cryo-EM as evidence. These structures were often validated using mass spectrometry or other biochemical tools. It is important to inform readers that if a protein is not visible in a complex structure, it does not necessarily mean the protein is absent or weakly bound. Instead, the protein could be flexible or unresolved due to technical limitations.

Secondly, the authors choose to focus on eS10, eS12, and eS31 located on the beak of 40S ribosomes in this review. However, empirically, the beak is just a morphological feature, and some researchers may also consider uS3 as a beak protein. The authors describe uS3 as a protein binding to the base of the beak. It would be helpful if the authors could clarify this point and provide an illustration of the location of uS3 in one of the structures.

Lastly, the authors focus on the compositional changes during the assembly/maturation of the beak structure. However, the beak RNA also undergoes significant changes based on available structures, and it would be useful to include an illustration of these changes.

Other minor comments:

Line 81: The PDB ID for 60S subunit in figure 1B needs to be corrected. 

Line 327: uS15 is probably a typo of uS19? 

Line 579: Emg1 should be corrected to Enp1? 

Author Response

  1. It is important to inform readers that if a protein is not visible in a complex structure, it does not necessarily mean the protein is absent or weakly bound. Instead, the protein could be flexible or unresolved due to technical limitations.

ANSWER: We agree with the reviewer. Indeed, this was already discussed in our text. Please, look at the third paragraph of the first heading "Introduction".

  1. It would be helpful if the authors could clarify this point and provide an illustration of the location of uS3 in one of the structures.

ANSWER: We thank the reviewer for her/his comment. We have labelled uS3 as situated at the base of the beak in the revised Figure 2 and modified the legend accordingly.

  1. Lastly, the authors focus on the compositional changes during the assembly/maturation of the beak structure. However, the beak RNA also undergoes significant changes based on available structures, and it would be useful to include an illustration of these changes.

ANSWER: We believe that this is a complicated issue, especially as there are controversies on the timing of assembly of some beak ribosomal proteins. We have included a sentence in our revised text to alert the readers of this circumstance.

  1. Other minor comments:

Line 81: The PDB ID for 60S subunit in figure 1B needs to be corrected.

ANSWER: The PDB ID 6OIG seems correct to us. It comes from reference Rai et al. RNA (2021), doi: 10.1261/rna.077610.120

Line 327: uS15 is probably a typo of uS19?

ANSWER: Thanks. Changed to uS19, which is encoded by the RPS15 gene.

Line 579: Emg1 should be corrected to Enp1?

ANSWER: We politely disagree. Emg1 is the RAF that is situated at the basal subdomain of the 90S particle and that interacts with the beak subdomain comprised of eS1, eS31 and Enp1. We have labelled Emg1 in the revised Figure 5 to make this point clearer.

Reviewer 2 Report

Comments and Suggestions for Authors

The structure and function of the ribosome and its evolution have been under intense investigation for decades. Despite a constant stream of new information,  ribosome assembly,  mechanism, and interaction with other cellular components are still not fully understood. Hence, it is important that investigators in the field periodically write reviews summarizing the status and progress of the relevant research. Jesus de la Cruz and co-authors have submitted a review focusing on a prominent substructure of the ribosome called the beak. The authors have collected a comprehensive collection of publications and presented the results with an insightful understanding of the functions of the beak. Thus, I have only a few comments for the authors to consider.

1.    The authors should specify what they consider “the ribosome core”. Yusupova and Yusopov presented the concept of ribosome core and shell, but do the authors refer to the core in this sense or in some other way? This becomes of interest in the discussion of the exchange of uL16, which is described as a core protein (line 117). A priori, it would seem to be difficult to extract a core protein from the mature ribosome, because of the surrounding shell However, according to ref 38 uL16/RPL10/Qsr1 is exchanged, even though this is not documented in ref 38, and only alluded to in the Trumpower paper referenced in ref 38. Inspecting the yeast ribosome structure the protein looks quite close to the ribosome surface. I apologize for being pedantic.

2.    Line 103: allowed whom?

3.    Line 282: Define subfigures in the legend.

Author Response

  1. The authors should specify what they consider “the ribosome core”. Yusupova and Yusopov presented the concept of ribosome core and shell, but do the authors refer to the core in this sense or in some other way? This becomes of interest in the discussion of the exchange of uL16, which is described as a core protein (line 117). A priori, it would seem to be difficult to extract a core protein from the mature ribosome, because of the surrounding shell However, according to ref 38 uL16/RPL10/Qsr1 is exchanged, even though this is not documented in ref 38, and only alluded to in the Trumpower paper referenced in ref 38. Inspecting the yeast ribosome structure the protein looks quite close to the ribosome surface. I apologize for being pedantic.

ANSWER: We thank the reviewer for this comment. Reference 38 shows how oxidized uL16 protein can be released from damage ribosomes by its dedicated chaperone Sqt1 in vitro. Since the exchangeability of uL16 is not further demonstrated in this article, we have softened the argument by using the adverb "potentially". Please see the revised text.

The concept of "ribosome core" is as defined by Yusupova and Yusupov; thus, it refers to the evolutionarily conserved common structure for ribosomes of bacteria, archaea and eukaryotes. As uL16 is a "universal protein", it thus belongs to the core. To clarify better this issue, we have modified the revised text to "uL16 is strategically positioned on the surface of the evolutionarily conserved core of the 60S r-subunit"

Line 103: allowed whom?

ANSWER: Rewritten: "allowed different authors to…"

Line 282: Define subfigures in the legend.

ANSWER: Figure 4 and its legend have been modified.